# Functional Characterisation of the Transcription Factor *GsWRKY23* Gene from *Glycine soja* in Overexpressed Soybean Composite Plants and *Arabidopsis* under Salt Stress

**DOI:** 10.3390/plants12173030

**Published:** 2023-08-23

**Authors:** Shile Sun, Xun Liu, Tianlei Zhang, Hao Yang, Bingjun Yu

**Affiliations:** 1Lab of Plant Stress Biology, College of Life Sciences, Nanjing Agricultural University, Nanjing 210095, China; 2College of Life Sciences, Xinjiang Agricultural University, Urumqi 830052, China

**Keywords:** *Glycine soja*, *GsWRKY23* gene, gene overexpression, soybean hairy root, transgenic *Arabidopsis*, salt tolerance

## Abstract

WRKY proteins are a superfamily of transcription factors (TFs) that play multiple roles in plants’ growth, development, and environmental stress response. In this study, a novel WRKY gene called *GsWRKY23* that is specifically upregulated in salt-tolerant *Glycine soja* accession BB52 seedlings was identified by transcriptomic analysis under salt stress. How the physiological functions and mechanisms of the *GsWRKY23* gene affect salt tolerance was investigated using transformations of soybean hairy roots and *Arabidopsis*, including wild-type (WT) and *atwrky23*-mutant plants. The results showed that *GsWRKY23* in the roots, stems, and leaves of BB52, along with its promoter in the cotyledons and root tips of *GsWRKY23pro*::GUS *Arabidopsis* seedlings, displayed enhanced induction under salt stress. GsWRKY23 localises to the nucleus and shows transcriptional activation ability in yeast cells. Compared to *GsWRKY23*-RNAi wild soybean hairy-root composite plants under salt stress, obvious improvements, such as superior growth appearance, plant height and fresh weight (FW), and leaf chlorophyll and relative water content (RWC), were displayed by *GsWRKY23*-overexpressing (OE) composite plants. Moreover, their relative electrolytic leakage (REL) values and malondialdehyde (MDA) contents in the roots and leaves declined significantly. Most of the contents of Na^+^ and Cl^−^ in the roots, stems, and leaves of *GsWRKY23*-OE plants decreased significantly, while the content of K^+^ in the roots increased, and the content of NO_3_^−^ displayed no obvious change. Ultimately, the Na^+^/K^+^ ratios of roots, stems, and leaves, along with the Cl^−^/NO_3_^−^ ratios of roots and stems, decreased significantly. In the transgenic WT-*GsWRKY23* and *atwrky23*-*GsWRKY23 Arabidopsis* seedlings, the salt-induced reduction in seed germination rate and seedling growth was markedly ameliorated; plant FW, leaf chlorophyll content, and RWC increased, and the REL value and MDA content in shoots decreased significantly. In addition, the accumulation of Na^+^ and Cl^−^ decreased, and the K^+^ and NO_3_^−^ levels increased markedly to maintain lower Na^+^/K^+^ and Cl^−^/NO_3_^−^ ratios in the roots and shoots. Taken together, these results highlight the role of *GsWRKY23* in regulating ionic homeostasis in NaCl-stressed overexpressed soybean composite plants and *Arabidopsis* seedlings to maintain lower Na^+^/K^+^ and Cl^−^/NO_3_^−^ ratios in the roots and shoots, thus conferring improved salt tolerance.

## 1. Introduction

Soil salinization is an increasingly serious problem worldwide, and the global salinized land area has reached approximately 1 billion hectares, accounting for 7% of the total land area [1]. In addition to natural soil salinity, climate change and human irrigation have also exacerbated soil salinization [2]. Soil salinity is one of the major environmental stresses that restrict plants’ growth and development, as well as crop yield. NaCl is the most soluble salt, with the highest content in soil [3]. When plants are exposed to a high-salt environment, plants will first be subjected to osmotic stress, and the lower water potential in the external soil will lead to a decrease in the water absorption capacity of the plants’ roots [4]. Second, the high concentration of Na^+^ will reduce the absorption of K^+^ and stimulate the outflow of K^+^, resulting in insufficient intracellular K^+^ concentration and inhibition of the enzyme catalysis controlled by K^+^, thus affecting normal metabolism in plants [5,6]. As anions such as Cl^−^, NO_3_^−^, and SO_4_^2−^ can be transported through nonselective anion channels or transports, excessive accumulation of Cl^−^ will lead to reduced absorption of anions such as NO_3_^−^ and SO_4_^2−^, resulting in the lack of nutrient elements such as N and S [7]. Therefore, Na^+^ and Cl^−^ constitute the main components of ionic toxicity for plants under salt-stressed environments. During their long evolutionary process, plants have formed complex and effective signal networks that can transform extracellular stress factors into intracellular signals and mediate plants’ response processes and adaptation mechanisms to various stresses by activating or inhibiting stress-related gene expression to improve adaptability to adverse environments. In this process, various transcription factors (TFs), which can be regulated as cascade switches, often play the role of upstream connection and coordination [8].

Transcription factors (TFs), also known as trans-acting factors, can activate or inhibit the transcription of downstream target genes by recognising *cis*-acting elements in the promoter of target genes, and they participate in a variety of biological processes in plants, including cell division, growth, development, ageing, and various abiotic or biotic stress responses [9]. When plants suffer from environmental stress, TFs, such as bZIP, NAC, MYB, DREB, WRKY, and other families, can activate or suppress the expression of target genes by binding to *cis*-acting elements, and then regulate related biochemical reactions and physiological metabolic processes through the expressed substances to reduce or avoid greater damage to plants. Regulating or suppressing the expression of genes encoding these transcription factors will generally improve the adaptability of plants to various stresses, such as salt, drought, low temperature, pathogens, and insects [10,11]. As one of the largest and most important TF families, the WRKY family derives its name from a highly conserved WRKY domain containing approximately 60 amino acid residues and a highly conserved WRKYGQK heptapeptide in its N-terminus, which is associated with DNA-binding activity. The C-terminus has a typical zinc-finger motif (C_2_H_2_ or C_2_HC type), which is involved in protein interaction and assisted DNA binding [12,13]. WRKY proteins can activate or suppress the transcription of target genes by specifically binding their domain to the T/CTGACC/T (W-box) *cis*-acting element in the target gene promoter. According to the number of WRKY domains and the type of zinc-finger motifs, WRKY proteins can be divided into three groups: group Ⅰ proteins contain two WRKY domains and a C_2_H_2_-type (or C_2_HC-type) zinc-finger motif; group Ⅱ proteins contain only one WRKY domain with a C_2_H_2_-type zinc-finger motif, which can be subdivided into five subgroups (Ⅱa-Ⅱe) according to the difference in amino acid sequence; group Ⅲ proteins consist of a WRKY domain and a zinc finger of the C_2_HC type [13]. Since *SPF1*, the first member of the *WRKY* gene family, was isolated from sweet potatoes [14], an increasing number of *WRKY* gene family members with multiple roles in plants have been identified. Many studies have shown that these *WRKY* genes mainly respond to biotic or abiotic stress [15,16] and play regulatory roles in leaf senescence [17], seed development [18], plant dormancy [19], and biosynthesis of secondary metabolites [20]. For example, the expression of *AtWRKY23* is controlled by auxin through the ARF7/19 transcriptional response pathway, which can stimulate flavonoid biosynthesis and is needed for proper root growth and embryo development [20]. WRKY23 from cucumber (*Cucumis sativus* L.), which is homologous to AtWRKY23, may be involved in the positive regulation of fruit peel yellowing [21].

Cultivated soybeans (*Glycine max*), as the main source of plant-derived protein, edible oil, and animal feed, play an important role in global agricultural production, sustainable development, and the human diet [22]. *G. max* was domesticated from its wild relative *G. soja* in China, and generally, *G. soja* displays relatively higher salt tolerance than *G. max* [23,24]. During long-term domestication and improvement, some key genes in the *G. soja* species for adaptation to stressful environments have been lost, resulting in obviously decreased genetic diversity in *G. max* cultivars [25,26]. Nevertheless, due to the lack of a reproductive barrier between *G. soja* and *G. max*, it is possible for these stress-tolerance-related genes in *G. soja* to be reintroduced into the cultivated soybeans through breeding approaches [27]. Therefore, it is of great importance to discover and identify the key salt-tolerance genes in *G. soja* for molecular design breeding of salt-tolerant *G. max* cultivars. In our previous study, transcriptomic analysis was carried out on the roots of halophytic *G. soja* BB52 seedlings under NaCl treatment to explore the molecular mechanisms of its strong salt tolerance. A large number of differentially expressed genes (DEGs) were enriched in Ca^2+^-signalling-related pathways in the plant-pathogen interaction pathway [25]. On this basis, according to the Ca^2+^ signal action on the MPK4 protein, the transcription factor *WRKY* gene family was enriched downstream of the MPK4 regulation pathway, and the transcription factor *GsWRKY23* gene, which was significantly upregulated in the roots of BB52 seedlings under salt stress, was identified. In this work, the function of the *GsWRKY23* gene in the salt tolerance of *G. soja* accession BB52 was investigated through multiple experimental techniques, including tobacco leaf transient expression, protein transcriptional activation assay and subcellular localisation, overexpression or RNA interference (RNAi) transformation of *G. soja* hairy roots mediated by *Agrobacterium rhizogenes*, and transgenic *Arabidopsis* (including WT and *atwrky23* mutant) mediated by *Agrobacterium tumefaciens*. The changes in the phenotype and physiological parameters of *GsWRKY23*-overexpressing/RNAi hairy-root soybean composite plants and transgenic *Arabidopsis* exposed to NaCl treatment were determined. The aims of this study were to reveal the physiological mechanisms of *GsWRKY23* mediating the alleviation of ionic toxicity in *G. soja* plants under salt stress, and to provide a crucial scientific basis and gene resources for molecular design breeding and salt-tolerance improvement in cultivated soybeans or other crops in the future.

## 2. Results

### 2.1. Discovery of the GsWRKY23 Gene and Amino Acid Sequence Analysis of Its Encoding Protein

In a previous study, Pi et al. [25] conducted a transcriptomic analysis of the roots of halophytic *G. soja* accession BB52 seedlings under NaCl treatment. On this basis, we screened 11 WRKY family members that play a role in regulating downstream gene expression involved in Ca^2+^-signalling-related pathways of the plant–pathogen interaction pathway, among which *GsWRKY23* was the member with the most salt-induced upregulation (Table 1 and Figure 1A). The coding DNA sequence (CDS) of the GsWRKY23 protein contained 320 amino acids, and a DNA-binding domain was identified at amino acid positions 170~211, including a WRKY domain and a C_2_H_2_ zinc-finger motif; thus, GsWRKY23 belonged to the WRKY Ⅱc subfamily. Amino acid sequence alignment showed that GsWRKY23 shared 99.6%, 93.1%, 91.2%, 74.8%, 67.9%, 63.7%, and 55.7% sequence identity with GmWRKY3, GmWRKY54, VuWRKY23, AhWRKY23, MtWRKY23, JrWRKY23, and AtWRKY23, respectively (Figure 1B). Further analysis using conserved domain alignments revealed that the WRKY domain was highly conserved, indicating that these proteins may actually have functions similar to GsWRKY23 (Figure 1C).

### 2.2. Response of GsWRKY23 and Its Promoter to Salt Stress

When 2-week-old wild soybean BB52 seedlings were exposed to 120 mM NaCl for 24 h, the expression levels of *GsWRKY23* in different tissues were analysed using qRT–PCR. At 0, 3, 6, 12, and 24 h of salt treatment, the expression patterns of *GsWRKY23* in roots, stems, and leaves first increased and then decreased. The upregulation multiples in roots, stems, and leaves reached maxima at 6 h of salt treatment, which were 5.71, 4.97, and 27.49 times, respectively (Figure 2A).

The promoter region (1632 bp) of *GsWRKY23* was inserted upstream of the *uidA* (*GUS*) gene of the PBI101 vector to obtain the recombinant reporter vector *GsWRKY23pro*::GUS, and *GsWRKY23pro*::GUS-transgenic *Arabidopsis* was then obtained by *A. tumefaciens* GV3101-mediated transformation (Appendix A). When treated with 120 mM NaCl for 24 h, obvious GUS staining (blue colour) was observed in both the cotyledons and root tips of transgenic *Arabidopsis* seedlings when compared to the untreated seedlings (Figure 2B). Thus, both the *GsWRKY23* gene and its promoter can respond to salt stress with an enhanced effect in *G. soja*.

### 2.3. Transcriptional Activation Ability and Subcellular Localisation of the GsWRKY23 Protein

The CDS of *GsWRKY23* was cloned into the pGBKT7 to construct a pGBKT7-*GsWRKY23* plasmid, and then pGBKT7 and pGBKT7-*GsWRKY23* were introduced into Y2HGold yeast cells to detect the transcriptional activation ability of the GsWRKY23 protein. As shown in Figure 3A, the yeastY2HGold strain and Y2HGold transformed with pGBKT7 or pGBKT7-*GsWRKY23* plasmids grew well on YPDA medium, but only the yeast strain transformed with the pGBKT7-*GsWRKY23* grew on SD/-Trp/-His medium and turned blue when X-α-Gal was added. This indicates that GsWRKY23 activated the expression of the reporter genes (*HIS* and *MEL1*) in yeast cells and showed transcriptional activation activity.

To explore the subcellular localisation of the GsWRKY23 protein, the CDS of *GsWRKY23* without a termination codon was cloned into the pCAMBIA1300-eGFP vector to obtain the 35S:: *GsWRKY23*::eGFP recombinant plasmid, and then the empty vector pCAMBIA1300-GFP (35S::eGFP) and the fusion vector 35S::*GsWRKY23*::eGFP were transferred into the young leaves of tobacco plants by *A. tumefaciens* GV3101-mediated transformation. As shown in Figure 3B by laser confocal microscopy, compared with the leaf cells transformed with 35S::eGFP, obvious green fluorescence signals were observed only in the cell nuclei of tobacco leaves injected with 35S::*GsWRKY23*::eGFP. These results may indicate that the GsWRKY23 protein was mainly located in the nucleus and functioned as a transcription factor.

### 2.4. Analysis of Salt Tolerance in GsWRKY23-Overexpressing and GsWRKY23-RNAi Wild Soybean Hairy-Root Composite Plants

Using the *GsWRKY23*-overexpressed vectors (*GsWRKY23*-OE) and RNA interference (*GsWRKY23*-RNAi) (Appendix A) and a hairy-root transformation system mediated by *A. rhizogenes*, empty vector (EV), *GsWRKY23*-OE, and *GsWRKY23*-RNAi soybean hairy-root composite plants were obtained (Figure 4A and Appendix A). Under normal conditions, there were no apparent differences in the growth phenotype, FW per plant, plant height, leaf chlorophyll content and RWC, REL values, and MDA contents in roots and leaves among the EV, *GsWRKY23*-OE, and *GsWRKY23*-RNAi soybean plants (Figure 5). When exposed to 120 mM NaCl solution for 3 d, the growth of all soybean plants was found to be significantly inhibited, while the growth of *GsWRKY23*-OE was observed to be relatively good. Compared with EV plants, FW per plant, plant height, leaf chlorophyll content, and RWC of *GsWRKY23*-OE plants were obviously increased, and the changes in plant height, REL value, and MDA content of *GsWRKY23*-OE plants reached significant levels (*p* ≤ 0.05). However, the salt damage to *GsWRKY23*-RNAi plants was seriously aggravated, with more stunted plant height, clearly decreased leaf chlorophyll content, and increased root and leaf REL values and MDA contents, especially in the aspect of leaf-related physiological indices, e.g., leaf chlorophyll content decreased by 22.22%, and leaf REL value and MDA content increased by 62.62% and 42.94%, respectively, which all reached significant levels in comparison with EV (*p* ≤ 0.05) (Figure 5).

Furthermore, when compared with the control, the contents of Na^+^ and Cl^−^ significantly increased in the roots, stems, and leaves of EV, *GsWRKY23*-OE, and *GsWRKY23*-RNAi wild soybean plants under NaCl stress, while the K^+^ content was significantly decreased and the NO_3_^−^ content showed no visible change. When compared with *GsWRKY23*-RNAi plants, the Na^+^ contents in the roots, stems, and leaves of OE plants decreased by 28.44%, 13.53%, and 4.86%, respectively, while the K^+^ contents increased by 8.35%, 21.73%, and 3.12%, respectively; correspondingly, the Na^+^/K^+^ ratios of roots, stems, and leaves were markedly decreased, by 33.89%, 28.87%, and 7.68%, respectively (*p* ≤ 0.05) (Figure 6A–C). Under NaCl treatment, and when compared with RNAi plants, the Cl^−^ contents in the roots and stems of OE plants decreased by 24.61% and 19.33%, respectively, which resulted in remarkable decreases in the Cl^−^/NO_3_^−^ ratio of roots and stems, by 21.52% and 26.88%, respectively (*p* ≤ 0.05) (Figure 6D–F).

Considering the ion-transporter-related genes as the pointcut, when *GsWRKY23*-OE and *GsWRKY23*-RNAi soybean composite seedlings were subjected to 120 mM NaCl solution for 6 h, compared with the control conditions, and except for the obvious downregulated expression of *GsNRT2*, the expression levels of *GsHKT1-1*, *GsSOS1*, *GsNHX2*, *GsNRT2*, *GsCLC1*, *GsCLC-b2*, and *GsCLC-c2* in roots were significantly upregulated under salt treatment. Among them, the expression levels of *GsHKT1-1* and *GsSOS1* were significantly upregulated in *GsWRKY23*-OE plants in contrast to EV plants but were clearly downregulated in *GsWRKY23*-RNAi plants, and *GsCLC*-c2 and *GsCLC*-b2 exhibited the opposite trend. In addition, there were no remarkable differences in the expression levels of *GsNHX2*, *GsNRT2*, and *GsCLC1* among the three soybean materials (Figure 4).

### 2.5. GsWRKY23-Transgenic Arabidopsis Exhibits Enhanced Salt Tolerance

After the *Arabidopsis atwrky23* mutant was confirmed by PCR identification (Appendix A), transgenic WT-*GsWRKY23* and *atwrky23*-*GsWRKY23* homozygous lines (T_3_) were generated in this study (Appendix A). When cultured on ½ MS medium, there was no visible variation in the seed germination rates of WT, *atwrky23*, WT-*GsWRKY23*, and *atwrky23*-*GsWRKY23*, all of which reached over 98% on the 7th day. When exposed to ½ MS medium plus 75 mM or 150 mM NaCl, the seed germination rates of the four *Arabidopsis* materials were all reduced, but WT-*GsWRKY23* and *atwrky23*-*GsWRKY23* displayed relatively higher germination rates (96.81% and 93.72%, respectively) than WT and *atwrky23* (81.92% and 75.0%, respectively) under 150 mM NaCl treatment on the 7th day (Figure 7A,B).

When the abovementioned 3-week-old *Arabidopsis* plants were treated with 120 mM NaCl for 7 d, their growth was markedly inhibited, and the WT and *atwrky23*—especially the mutant—suffered the most damage, while the WT-*GsWRKY23* and *atwrky23*-*GsWRKY23* showed a relatively better growth phenotype (Figure 7C). Compared with WT plants under salt stress, the FW per plant, leaf chlorophyll content, and RWC of *atwrky23* were significantly decreased, while the REL value and MDA content in shoots were significantly increased. When the *GsWRKY23* gene was transformed into *Arabidopsis* WT and the *atwrky23* mutant, the salt tolerance of WT-*GsWRKY23* and *atwrky23-GsWRKY23* was enhanced, and this was simultaneously accompanied by the obvious recovery of FW per plant, leaf chlorophyll content, and RWC, as well as a significant decrease in REL value and MDA content in the shoots (*p* ≤ 0.05), and the mitigation effect of salt damage on *atwrky23-GsWRKY23* was more obvious (Figure 7D–H).

Further analysis of the contents of Na^+^, K^+^, Cl^−^, and NO_3_^−^, as well as the ratios of Na^+^/K^+^ and Cl^−^/NO_3_^−^, in the roots and shoots of WT, *atwrky23*, WT-*GsWRKY23*, and *atwrky23*-*GsWRKY23* plants under salt stress showed that, compared with the control, salt treatment significantly increased the contents of Na^+^ and Cl^−^ in the roots and shoots of the above four *Arabidopsis* plants, especially in the shoots; for example, the contents of Na^+^ and Cl^−^ in WT increased by 10.71 and 5.47 times, respectively, and those of *atwrky23* increased by 16.33 and 6.76 times, respectively. However, the contents of Na^+^ and Cl^−^ in the shoots of WT-*GsWRKY23* and *atwrky23-GsWRKY23* only increased by 4.29, 2.54 and 4.97, 3.07 times, respectively (Figure 8A,D).

Under salt stress, the contents of K^+^ and NO_3_^−^ in the roots and shoots of WT and *atwrky23* declined significantly, but the decrease in K^+^ content in the roots and shoots of WT-*GsWRKY23* and *atwrky23-GsWRKY23* was obviously lower. Furthermore, the NO_3_^−^ contents in the roots and shoots increased significantly compared with those of WT and *atwrky23*. Among them, the increase in NO_3_^−^ content in the shoots was larger, increasing by 46.12% and 77.80% compared with WT and *atwrky23*, respectively, which already exceeded the control level (*p* ≤ 0.05) (Figure 8B,E).

The above changes resulted in the ratios of Na^+^/K^+^ and Cl^−^/NO_3_^−^ in the roots and shoots of WT and *atwrky23* plants under salt stress being significantly higher than the control levels, and the increases in *atwrky23* plants were greater. The ratios of Na^+^/K^+^ and Cl^−^/NO_3_^−^ in roots increased by 8.16 and 12.41 times, respectively, while those in shoots increased by 30.02 and 10.85 times, respectively. Under the transformation of the *GsWRKY23* gene into *Arabidopsis* WT and the *atwrky23* mutant, the ratios of Na^+^/K^+^ and Cl^−^/NO_3_^−^ in the roots and shoots of WT-*GsWRKY23* and *atwrky23-GsWRKY23* plants decreased significantly (*p* ≤ 0.05), especially in *atwrky23-GsWRKY23* plants, whose ratios of Na^+^/K^+^ and Cl^−^/NO_3_^−^ in roots decreased by 69.42% and 72.88%, respectively, while those in the shoots decreased by 80.89% and 75.10%, respectively (Figure 8C,F).

## 3. Discussion

### 3.1. Identification and Characteristics of the Transcription Factor GsWRKY23 Gene from the Salt-Tolerant G. soja Accession BB52

Under saline environments, a great amount of Na^+^ and Cl^-^ enters plants’ cells, and the cellular absorption of K^+^ and NO_3_^−^ clearly declines, leading to sharp rises in the Na^+^/K^+^ and Cl^−^/NO_3_^−^ ratios in plants, disrupted ionic homeostasis, and a series of physiological metabolism disorders and final plant growth inhibition [28]. The salt-borne *G. soja* accession BB52 was discovered and collected from the coastal area in Shandong Province, China, and many years of research have suggested that the BB52 accession used in this work possesses a higher salt tolerance than other *G. soja* species (such as N23227) and *G. max* cultivars (such as Jackson and N23674) [23,24,29]. Compared with the *G. max* cv. N23674, *G. soja* BB52 materials display stronger salt tolerance, which is mainly related to their superior ability to accumulate the absorbed Na^+^ and Cl^−^ in the roots and reduce the transport and distribution to the stems and leaves under salt stress [23,24,25]. Furthermore, maintaining anion homeostasis by maintaining relatively high NO_3_^−^ levels in the stems and leaves of soybean plants to reduce the Cl^−^/NO_3_^−^ ratio under salt treatment is also attributed to their stronger salt tolerance [24,29].

In the present study, previous transcriptomic data were used to analyse the differentially expressed genes (DEGs) in the roots of salt-stressed BB52 seedlings, and the significantly upregulated *GsWRKY23* was screened downstream of the MPK4 regulation pathway (Figure 1A). This is consistent with the tissue expression patterns of the *GsWRKY23* gene in BB52 seedlings observed during the 24 h of 120 mM NaCl treatment (Figure 2A), and it may also indicate enhanced induction of the *GsWRKY23* gene under NaCl treatment. Meng et al. found that specifically upregulated genes in salt-tolerant plant species may play more positive roles in salt stress response and adaptation [30]. The deeper GUS staining of the transgenic *GsWRKY23pro*::GUS *Arabidopsis* seedlings (Figure 2B) may suggest that the promoter of the *GsWRKY23* gene also displayed an enhanced response to NaCl stress. Moreover, obvious transcriptional activation activity of GsWRKY23 was verified using a yeast system (Figure 3A), and transient transformation of tobacco indicated that GsWRKY23 was expressed in the nucleus (Figure 3B), which indicates that GsWRKY23 is a nucleus-localised protein and functions as a transcription factor.

### 3.2. Mechanisms of the GsWRKY23 Gene Conferred Salt Tolerance to Overexpressed Soybean Composite Plants and Arabidopsis

To date, the *WRKY23* gene and its homologues have been reported in plants such as *Arabidopsis thaliana*, *Cucumis sativus*, *Populus tremula*, *Oryza sativa*, and *G. max*, and this suggests biological functions involved in the responses of plants to stressful environments. For example, overexpression of *AtWRKY30* enhanced the seed germination rate of transgenic *Arabidopsis thaliana* under salt and oxidative stresses [31]. *OsWRKY42*, as a negative regulator of jasmonic acid (JA)-mediated plant stress responses, conferred enhanced salt tolerance to *OsWRKY42*-transgenic *Arabidopsis*, which was mainly related to inducing anthocyanin production and reducing ROS levels [32]. In *G. max*, Li et al. found that *GmWRKY45*-transgenic *Arabidopsis* plants displayed a significantly higher survival rate and soluble sugar content and, thus, improved salt tolerance compared to WT plants [33]. Zhou et al. found that *GmWRKY21*-transgenic *Arabidopsis* plants were tolerant to cold stress, whereas *GmWRKY54* conferred salt and drought tolerance, and *GmWRKY13*-transgenic *Arabidopsis* seedlings were more sensitive to NaCl and mannitol stress [34]. Bao et al. found that overexpression of *PcWRKY33* from *Polygonum cuspidatum* increased salt sensitivity in transgenic *Arabidopsis* seedlings by inhibiting the increased expression of *AtSOS1*, *AtSOS2*, *AtSOS3*, and *AtNHX1* under salt treatment, accumulating excessive Na^+^ and thereafter increasing the Na^+^/K^+^ ratio [35]. Song et al. reported that SbWRKY50, a negative transcription factor of the salt stress response in sweet sorghum, enhanced salt sensitivity in transgenic *Arabidopsis* by binding to the W-box in the upstream promoter of *AtSOS1* and, accordingly, significantly increased the Na^+^ content and Na^+^/K^+^ ratio in roots [36]. Currently, there are few reports on the *G. soja* transcription factor WRKY gene family, such as *GsWRKY15a*, which is involved in the control of seed size in wild soybeans and was significantly more weakly expressed than its orthologous *GmWRKY15a* in *G. max* during pod development [37]. Compared to WT, *GsWRKY20*-transgenic *Arabidopsis* displayed enhanced drought tolerance, which was associated with increased susceptibility to ABA and facilitation of ABA-induced stomatal closure [38]. Furthermore, *GsWRKY20*-overexpressing alfalfa showed increased salt tolerance as a result of accumulating less Na^+^ and more K^+^ in both leaves and roots [39], and *GsWRKY20*-overexpressing soybeans displayed strongly increased drought tolerance, with lower stomatal density and faster stomatal closure [40].

However, there have been no reports on the *GsWRKY23* gene of wild soybeans and its related functions under salt stress or other stressful conditions. Therefore, the physiological functions of *GsWRKY23* in the responses of overexpressed hairy-root composite soybean plants and transgenic *Arabidopsis* to NaCl treatment were investigated for the first time in this study, to further decipher the molecular mechanisms of the *G. soja* BB52 accession with strong salt tolerance in terms of transcription factors regulating functional gene expression. In this work, the Na^+^/K^+^ ratios in the roots, stems, and leaves of *GsWRKY23*-OE soybean hairy-root composite plants, as well as Cl^−^/NO_3_^−^ in roots and stems, were significantly lower than those of *GsWRKY23*-RNAi plants under NaCl treatment (Figure 6C,F). These findings are also consistent with the enhanced salt tolerance phenotype (Figure 5A–C) or mitigated salt injury indices (Figure 5D–G) of *GsWRKY23*-OE plants compared to *GsWRKY23*-RNAi plants. Du et al. reported that *RtWRKY1*-transgenic *Arabidopsis* displayed enhanced salt stress tolerance in addition to exhibiting lower MDA content, Na^+^ content, and Na^+^/K^+^ ratio than WT plants under NaCl treatment, resulting in upregulated *AtSOS1* expression, but no effect on the expression level of *AtNHX1* was observed in response to NaCl stress [41]. *OsWRKY50*-overexpressing rice plants displayed increased tolerance to salt stress compared to WT plants by improving the expression of *OsLEA3*, *OsRAB21*, *OsHKT1;5*, and *OsP5CS1* under NaCl stress [16]. In our study, we also found that the expression levels of *GsHKT1-1* and *GsSOS1* were significantly upregulated in the roots of NaCl-treated *GsWRKY23*-OE plants in contrast to EV plants, but they were clearly downregulated in *GsWRKY23*-RNAi plants (Figure 4B,C), which may be causally related to the significantly lower Na^+^/K^+^ ratios in the roots, stems, and leaves of salt-stressed *GsWRKY23*-OE plants compared to those of *GsWRKY23*-RNAi plants. However, no similar situation was found for the other ion-transporter-related coding genes, such as *GsNHX2*, *GsNRT2*, *GsCLC1*, *GsCLC-b2*, and *GsCLC-c2*; in particular, the performance of *GsCLC-b2* and *GsCLC-c2* (their encoded proteins are generally thought to be related to the accumulation of anions such as Cl^−^ and NO_3_^−^ in the cell vacuoles of plants under salt stress [24,29]) was opposite to that of *GsHKT1-1* and *GsSOS1* (Figure 4D–H). When the *GsWRKY23* gene was transferred into *Arabidopsis* WT plants and *atwrky23* mutants, both transgenic plants displayed alleviated salt injury, especially *atwrky23*-*GsWRKY23* (Figure 7C). Correspondingly, the ratios of Na^+^/K^+^ and Cl^−^/NO_3_^−^ in the shoots and roots of WT-*GsWRKY23* and *atwrky23*-*GsWRKY23* were significantly lower than those of WT and *atwrky23* (Figure 8C,F). Overexpression of *GsWRKY23* also enhanced the expression of *AtSOS1* and *AtHKT1* in the shoots of transgenic *Arabidopsis* WT and *atwrky23* seedlings under salt treatment (Appendix A). These results indicate that *GsWRKY23* is involved in alleviating the ionic toxicity of overexpressed soybean and *Arabidopsis* plants under salt stress by regulating ionic uptake and transport in plants to adjust the ratios of Na^+^/K^+^ and Cl^−^/NO_3_^−^ for ionic homeostasis. Alabd et al. reported that PpABF3 could target the G-box *cis*-element in the promoter of *PpWRKY44*, and PpWRKY44 directly bound to a W-box on the promoter of *PpALMT9* to activate its expression and, thus, was positively involved in salinity-induced malate accumulation and pear fruit quality [42]. However, little is known about the downstream target genes and/or interacting proteins of GsWRKY23. Further studies are needed to identify these genes and/or interacting proteins, and to better understand how GsWRKY23 (alone or together with its interacting proteins) regulates target gene expression and functions for maintaining ionic homeostasis and enhancing antioxidant capacity to alleviate salt damage or attribute strong salt tolerance to *G. soja* BB52 plants. In addition, considering that GmWRKY3 and GsWRKY23 possess highly similar amino acid sequences (Figure 1B,C), joint exploration of the role of *GmWRKY3* may better assist us in fully revealing the salt tolerance function of *GsWRKY23*.

In conclusion, the transcription factor *GsWRKY23* gene, which was significantly upregulated under NaCl stress, was screened for the first time from the genome of the salt-tolerant *G. soja* BB52 accession. Both the *GsWRKY23* gene and its promoter displayed an enhanced response to NaCl stress. GsWRKY23 is a nucleus-localised protein that functions as a transcription factor. Analysis of phenotypic and physiological indices of EV, *GsWRKY23*-OE, and *GsWRKY23*-RNAi hairy-root composite soybean plants under salt stress indicated that *GsWRKY23* played a positive role in the process of salt stress response and adaptation, which could reduce the contents of Na^+^ and Cl^−^ in plants and increase their K^+^ content to maintain lower Na^+^/K^+^ and Cl^−^/NO_3_^−^ ratios, along with ionic homeostasis for contributing to stronger salt tolerance. These similar effects were further verified in *GsWRKY23*-transgenic *Arabidopsis*. These findings not only provide an important research basis for further revealing the molecular functions of the *GsWRKY23* gene and the mechanisms underlying the strong salt tolerance of *G. soja* BB52 materials, but also provide a valuable theoretical basis and gene resources for molecular genetic improvement and germplasm innovation of salt-tolerant soybean cultivars and other crops.

## 4. Materials and Methods

### 4.1. Plant Materials, Bacterial Strains, and Plasmids

Plant seeds included *Glycine soja* BB52 accessions, tobacco (*Nicotiana benthamiana* LAB), *Arabidopsis thaliana* (Col-0 wild type, WT), and *atwrky23* mutants (SALK_006855C). The bacterial strains included *Escherichia coli* DH5α, *A. tumefaciens* strain GV3101, and *A. rhizogenes* strain K599. The yeast strain Y2HGold, yeast expression vector pGBKT7, and plant binary vectors pCAMBIA1300-GFP, PBI101, and pFGC5941 were used in this study.

### 4.2. Phylogenetic Analysis and Multiple Alignment of Amino Acid Sequences of the WRKY Family

The amino acid sequence of GsWRKY23 (Glysoja.02G002639) was obtained from the wild soybean database (http://www.wildsoydb.org/Gsoja_W05/ (accessed on 5 June 2023)), and amino acid sequences of the WRKY23 family members with high homology to GsWRKY23 were retrieved and downloaded from the NCBI database (https://www.ncbi.nlm.nih.gov/ (accessed on 6 June 2023)). The phylogenetic tree of the WRKY family was constructed using the neighbour-jointing (NJ) method with MEGA X software, and the amino acid sequence alignment of WRKY family members was performed using DNAMAN 8 software.

### 4.3. Soybean Plant Culture, Salt Treatment, RNA Extraction, and qRT–PCR Analysis

Seeds of *G. soja* BB52 were surface-sterilised with 75% ethanol, soaked with deionised water for 12 h, and germinated in the dark at 25 °C. Then, the germinated seeds were grown in plastic pots containing quartz sand fertigated with ½-strength Hoagland nutrient solution, and then maintained in a greenhouse at 22~25 °C with a 16 h light/8 h dark photoperiod, light irradiation of approximately 100 μmol m^−2^ s^−1^, and 60~70% relative humidity. When the first trifoliate leaves were expanded, the seedlings were cultured with ½-strength Hoagland solution containing 120 mM NaCl for 0, 3, 6, 12, and 24 h, and then the roots, stems, and leaves were sampled, immediately frozen in liquid nitrogen, and stored at −80 °C. Total RNA was isolated from plants using the Spin Column Plant Total RNA Purification Kit (Sangon Biotech, Shanghai, China). The reverse transcription was performed using Hifair II 1st Strand cDNA Synthesis SuperMix (Yeasen, Shanghai, China). Quantitative real-time PCR (qRT–PCR) was performed on an ABI 7500 fluorescence quantitative PCR instrument (Thermo Fisher Scientific China, Inc., Shanghai, China) using the 2 × Hieff qPCR SYBR Green Master Mix (Low Rox Plus) (Yeasen). Three independent biological replicates and three independent technical replicates were included for each sample. The soybean *UBI3* gene (LOC114401261) was used as an internal control, and the relative expression was calculated using the 2^−ΔΔCt^ method [43]. All of the primers used for qRT–PCR are listed in Appendix A.

### 4.4. Expression Vector Construction

The promoter (1632 bp upstream of ATG) of *GsWRKY23* was inserted upstream of the *GUS* (*uidA*) gene of the PBI101 vector to obtain the recombinant vector *GsWRKY23pro*::GUS. The CDS of *GsWRKY23* was amplified by PCR using the specific primers (Appendix A) and inserted into the yeast expression vectors pGBKT7, plant binary vectors pCAMBIA1300-GFP, and pFGC5941, to generate the fusion constructs pGBKT7-*GsWRKY23*, 35S::*GsWRKY23*::eGFP (without a stop codon), and 35S::*GsWRKY23* overexpression (OE) vector. A 200 bp fragment was amplified from the CDS of *GsWRKY23* as the RNAi target and inserted upstream and downstream of the CHSA intron of the pFGC5941 vector in forward and reverse orientations, respectively, to obtain the *GsWRKY23*-RNAi vector. The CDS of *OsMADS* (encoding a nuclear locator protein as a label) without a stop codon was inserted into the pCX-DR-RFP vector to obtaine the RFP fusion construct 35S::*OsMADS*::RFP.

### 4.5. GUS Staining of Transgenic Arabidopsis

The fusion construct *GsWRKY23pro*::GUS was transformed into *Arabidopsis* Col-0 plants using the *A. tumefaciens* strain GV3101-mediated floral dip method [44]. Ten-day-old transgenic *Arabidopsis* seedlings planted on ½ MS agar medium were stressed with 120 mM NaCl solution for 24 h, and the untreated seedlings were used as a control. Then, these seedlings were sampled for the GUS staining assay as described by Jefferson et al. [45].

### 4.6. Transcriptional Activation Assay of GsWRKY23 in Yeast

A transcriptional activation assay of GsWRKY23 was conducted as described by Liu et al. [46]. Briefly, the fusion plasmid pGBKT7-*GsWRKY23* was introduced into the yeast strain Y2HGold, and the empty vector pGBKT7 was used as a negative control. The strains carrying pGBKT7 or pGBKT7-*GsWRKY23* and the Y2H strain were inoculated on YPDA, SD/-Trp/-His, and SD/-Trp/-His/X-α-Gal media to observe the growth of the strains and identify the transcriptional activation of GsWRKY23.

### 4.7. Subcellular Localisation Analysis of GsWRKY23

The fusion constructs 35S::*GsWRKY23*::GFP and 35S::*OsMADS*::RFP were transformed into the *A. tumefaciens* strain GV3101. Leaves of four-week-old tobacco plants were co-transformed with the fusion expression vector 35S::*GsWRKY23*::eGFP or 35S::*OsMADS*::RFP as described by He et al. [11], and the empty vector 35S::eGFP was used as a control. Then, the injected tobacco plants were cultured normally in a greenhouse for 24~48 h. Then, the GFP and RFP signals were visualised under a laser confocal fluorescence microscope (Ultra VIEW VoX, PerkinElmer, Waltham, MA, USA).

### 4.8. Salt-Tolerance Assays of GsWRKY23-OE and -RNAi Soybean Hairy-Root Composite Plants

The 35S::*GsWRKY23*-OE vector, -RNAi vector, and empty vector (pFGC5941) were transformed into *A. rhizogenes* K599, and soybean hairy-root transformation was mediated according to the methods described in our previous work [47]. Positive transgenic soybean hairy-root composite plants were identified by PCR, and then they were transferred to ½-strength Hoagland solution with or without 120 mM NaCl, and the plants transformed with an empty vector (EV) were used as negative controls. After 6 h, total RNA was extracted from the hairy roots of plants under control and NaCl treatments. Using the same method as previously described, qRT–PCR was used to detect the expression level changes of *GsWRKY23* and the ion-transporter-related genes (such as *GsHKT1-1*, *GsSOS1*, *GsNHX2*, *GsNRT2*, *GsCLC1*, *GsCLC-b2*, and *GsCLC-c2*) in EV, *GsWRKY23*-OE, and *GsWRKY23*-RNAi plants. Three independent biological replicates and three independent technical replicates were included for each sample. The specific primers used for qRT–PCR are also listed in Appendix A. After 3 d, phenotypic analysis of EV, *GsWRKY23*-OE, and *GsWRKY23*-RNAi plants was performed. Physiological parameters such as FW per plant, plant height, leaf chlorophyll content and RWC, REL values, and MDA contents in the roots and leaves were measured as described in our previous work [48]. The roots, stems, and leaves were sampled, dried, and ground into powder for the assays of the contents of Na^+^, K^+^, Cl^−^, and NO_3_^−^ according to the method described by Zhou and Yu [49].

### 4.9. Salt-Tolerance Analysis of GsWRKY23-Transgenic Arabidopsis WT and Mutant Plants

The *Arabidopsis atwrky23* mutant was obtained from the Arashare platform, and homozygous mutants were screened using the three-primer method. The vector *GsWRKY23*-OE was transformed into *Arabidopsis* WT and mutant plants using the floral dip method [44]. Homozygous T_3_ lines of WT-*GsWRKY23* (such as L1, L2, L3) and *atwrky23*-*GsWRKY23* (such as l1, l2, and l3) were obtained. The seeds of WT, *atwrky23*, WT-*GsWRKY23*, and *atwrky23*-*GsWRKY23* were sterilised with 75% ethanol and sown on ^1^/_2_ MS containing 0 (control), 75, and 150 mM NaCl, kept at 4 °C for 2 d and cultured horizontally in a greenhouse. Germination rates were calculated daily and photographed on the 7th day. Furthermore, seeds of the WT, *atwrky23*, and transgenic lines were sown in pots filled with peat moss and vermiculite (1:1). Three-week-old *Arabidopsis* plants were stressed with 120 mM NaCl for 7 d for phenotypic analysis, and the untreated plants were used as controls. Measurements of FW per plant, leaf chlorophyll content and RWC, REL values, and MDA contents in the shoots were also conducted. Finally, the shoots and roots were sampled, dried, and ground into powder for the assays of the contents of Na^+^, K^+^, Cl^−^, and NO_3_^−^.

### 4.10. Data Analysis

Data were analysed and presented as the mean ± SD for each treatment (*n* = 3, except for FW per plant and plant height, where *n* = 6, and seed germination of *Arabidopsis* where *n* = 32) using SPSS software (version 20.0). Significant differences among means were determined by Duncan’s test at the *p* ≤ 0.05 significance level.

## Figures and Tables

**Figure 1 plants-12-03030-f001:**
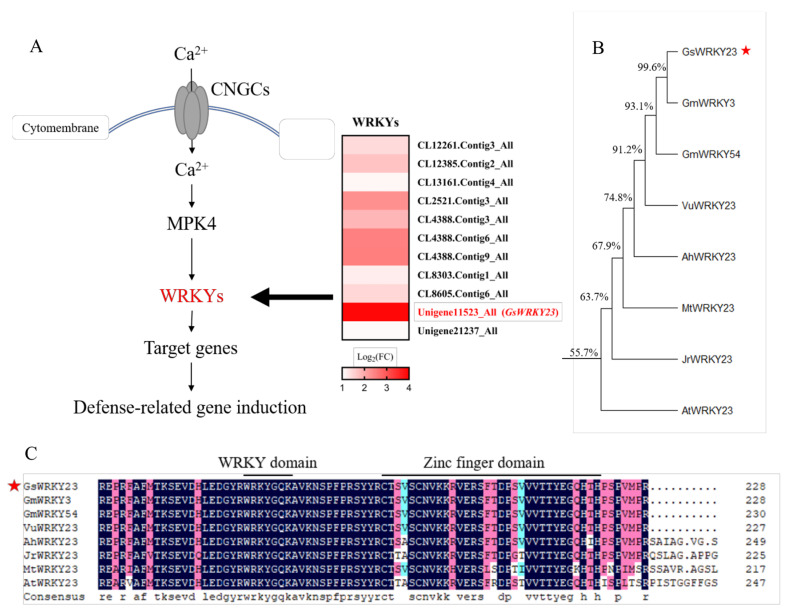
Screening of *GsWRKY23* and its encoding protein amino acid sequence analysis: (**A**) Screening the Ca^2+^-signal-related pathway of plant–pathogen interaction and the expression change heatmap of enriched WRKY transcription factors in the transcriptome data. (**B**) Phylogenetic analysis of GsWRKY23 and the closely related WRKY protein in other plant species. The selected WRKYs are GmWRKY3 (AJB84597.1) and GmWRKY54 (NP_001237438.1) from *Glycine max*, VuWRKY23 (XP_047168319.1) from *Vigna umbellate*, AhWRKY23 (XP_029144762.1) from *Arachis hypogaea*, JrWRKY23 (XP_018824247.2) from *Juglans regia*, MtWRKY23 (XP_003625994.2) from *Medicago truncatula*, and AtWRKY23 (NP_182248.1) from *Arabidopsis thaliana*. (**C**) Alignment of the amino acid sequence of GsWRKY23 with its homologous proteins. Identical amino acids are indicated with black shadowing and GsWRKY23 is indicated with red star. Features of the sequence include a WRKY domain (WRKYGQK) and a C_2_H_2_ zinc-finger domain (both highlighted with the bold black lines above the alignment).

**Figure 2 plants-12-03030-f002:**
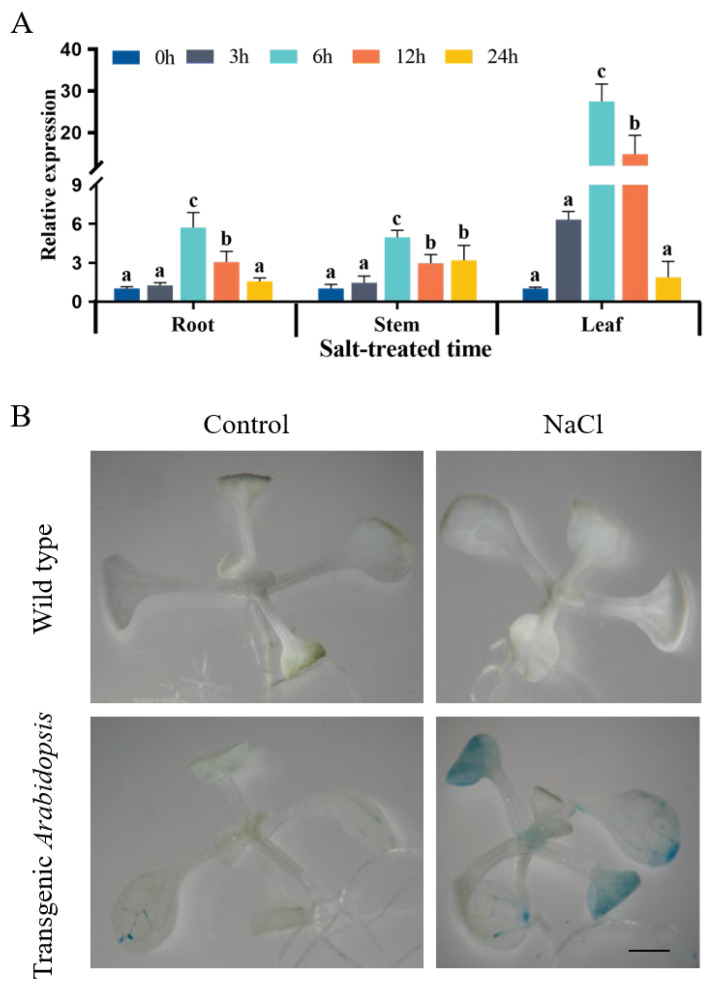
Tissue expression patterns of *GsWRKY23* gene in wild BB52 soybean seedlings during 24 h of 120 mM NaCl treatment (**A**), and GUS staining of wild-type and *GsWRKY23pro*::GUS *Arabidopsis* seedlings under 120 mM NaCl for 24 h; bar = 3 mm (**B**); different letters indicate groups with statistically significant differences using Duncan’s test (*p* ≤ 0.05).

**Figure 3 plants-12-03030-f003:**
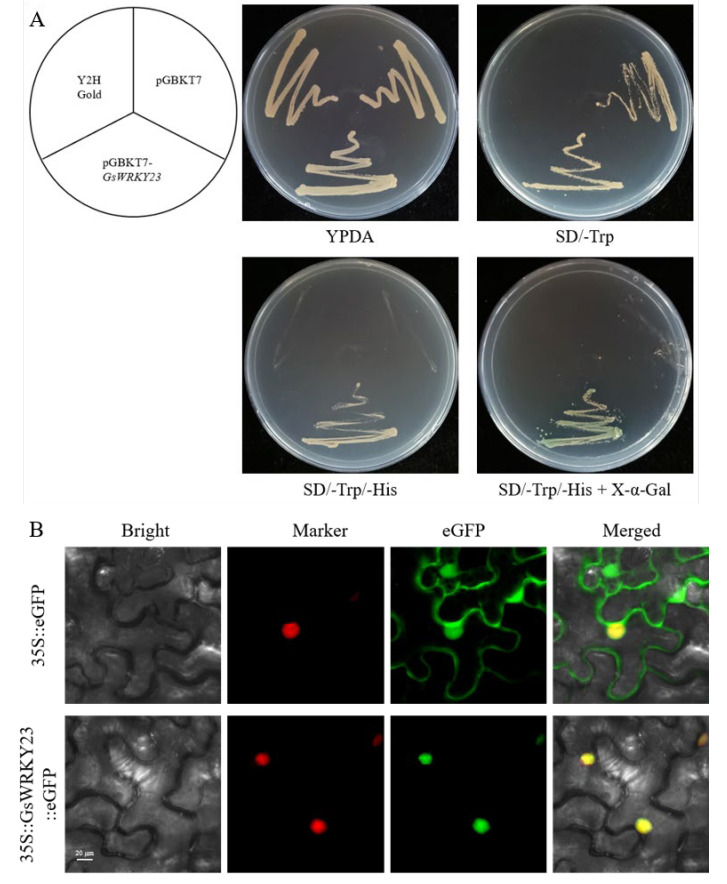
(**A**) Transcriptional activation ability of GsWRKY23 in the yeast strain Y2HGold. (SD/-Trp) indicates selective medium lacking Trp, (SD/-Trp/-His) indicates selective medium lacking Trp and His, (SD/-Trp/-His/X-α-Gal) indicates selective medium lacking Trp, His, and plus 20 mM X-α-Gal. (**B**) Subcellular localisation of GsWRKY23 in tobacco leaf epidermal cells; empty vector 35S::eGFP as a control; OsMADS3 as a nucleus marker protein; bar = 20 µm.

**Figure 4 plants-12-03030-f004:**
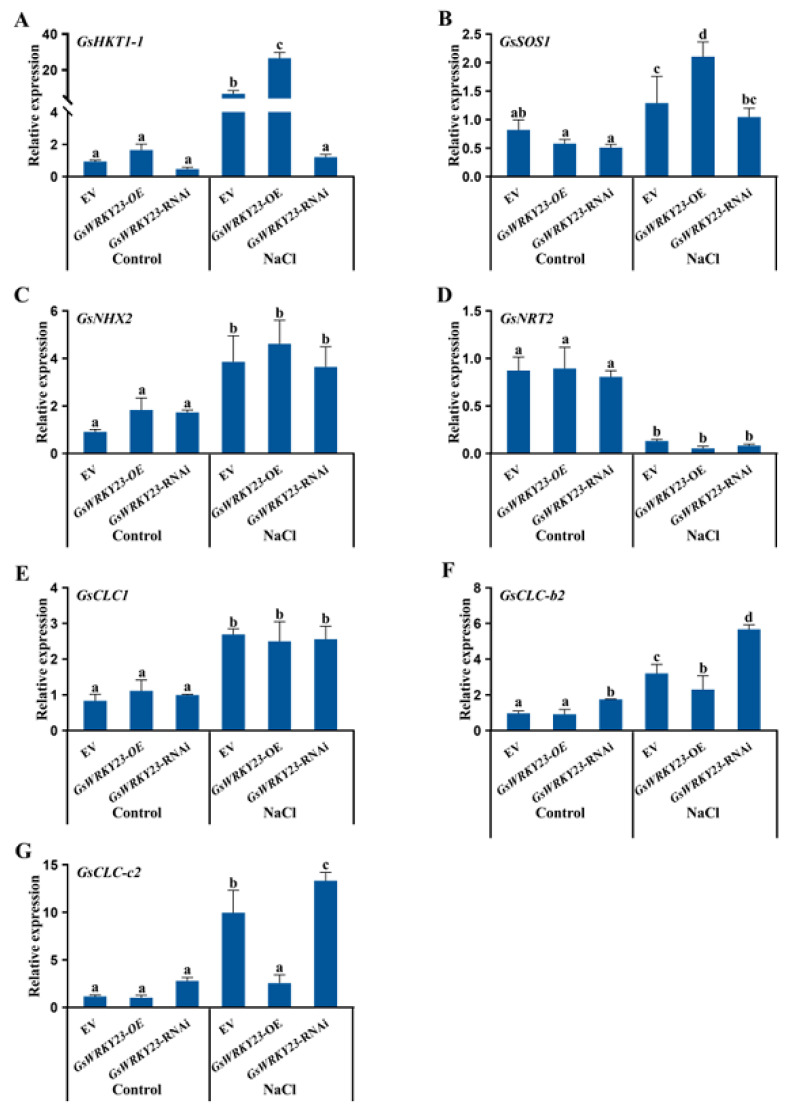
Expression levels of the ion-transporter-related genes in the roots of *GsWRKY23*-OE and *GsWRKY23*-RNAi hairy-root composite soybean plants under 120 mM NaCl solutions for 6 h: (**A**) *GsHKT1-1*, (**B**) *GsSOS1*, (**C**) *GsNHX2*, (**D**) *GsNRT2*, (**E**) *GsCLC1*, (**F**) *GsCLC-b2*, (**G**) *GsCLC-c2*. Means ± SD are shown (*n* = 3); different letters indicate groups with statistically significant differences using Duncan’s test (*p* ≤ 0.05).

**Figure 5 plants-12-03030-f005:**
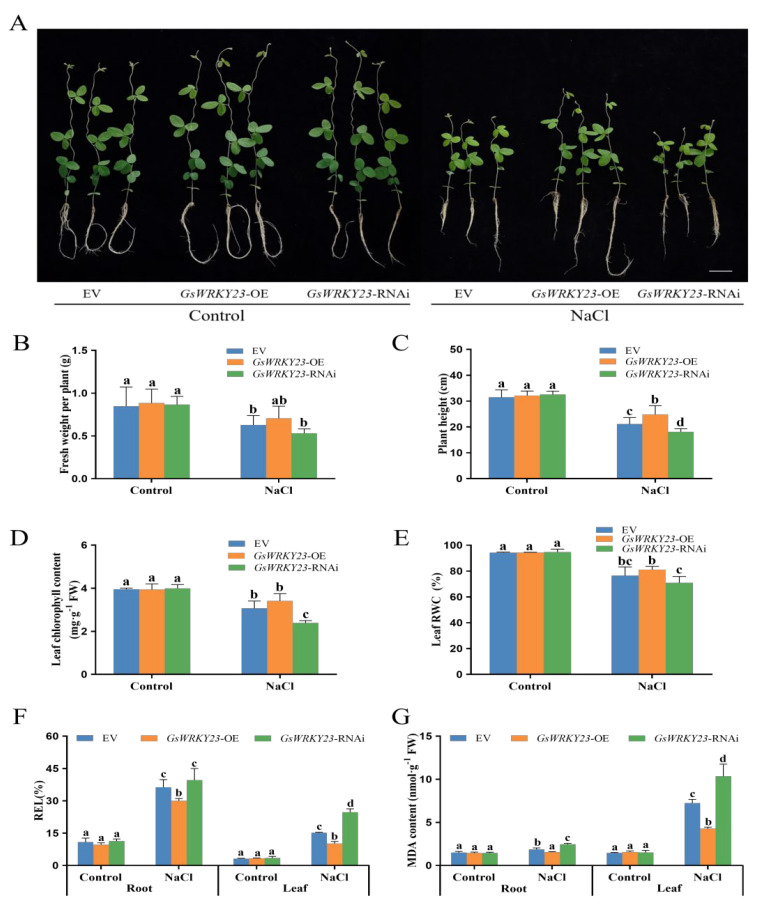
Analysis of salt tolerance of *GsWRKY23*−OE and *GsWRKY23*−RNAi soybean hairy−root composite plants: (**A**) Growth phenotype under 120 mM NaCl solutions for 3 d; bar = 5 cm. (**B**) Fresh weight (FW) per plant. (**C**) Plant height. (**D**) Leaf chlorophyll content. (**E**) Leaf RWC. (**F**) REL value. (**G**) MDA content. Means ± SD are shown (*n* = 3, except for the measurement of plant height and FW per plant, where *n* = 6); different letters indicate groups with statistically significant differences using Duncan’s test (*p* ≤ 0.05).

**Figure 6 plants-12-03030-f006:**
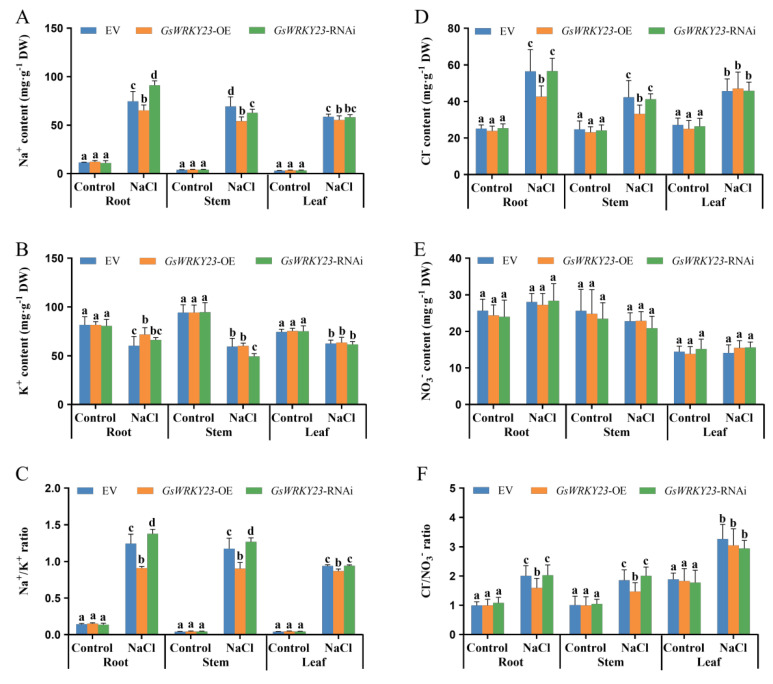
Ion contents in the roots, stems, and leaves of *GsWRKY23*−OE and *GsWRKY23*−RNAi hairy−root composite soybean plants under 120 mM NaCl solutions for 3 d: (**A**) Na^+^ content, (**B**) K^+^ content, and (**C**) Na^+^/K^+^ ratio. (**D**) Cl^−^ content, (**E**) NO_3_^−^ content, and (**F**) Cl^−^/NO_3_^−^ ratio. Means ± SD are shown (*n* = 3); different letters indicate groups with statistically significant differences using Duncan’s test (*p* ≤ 0.05).

**Figure 7 plants-12-03030-f007:**
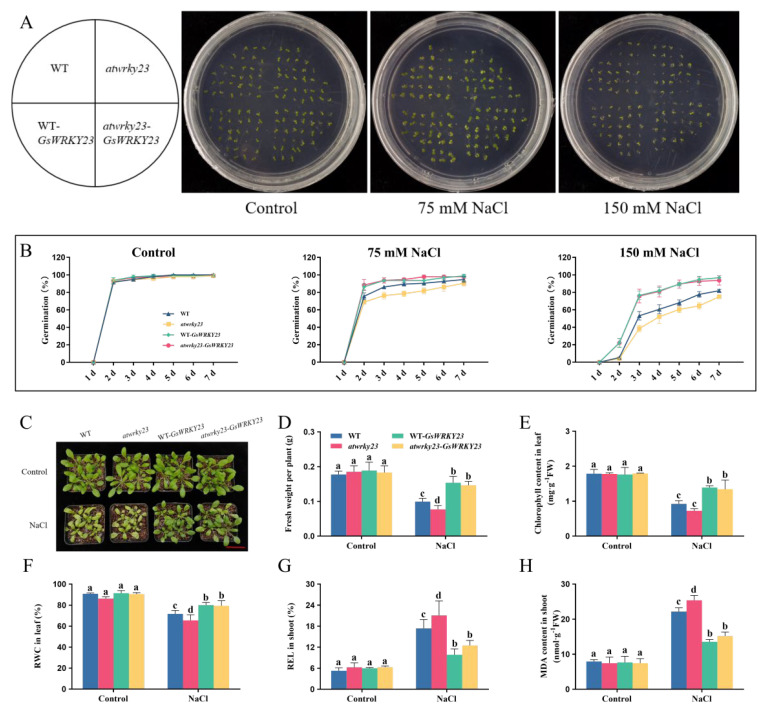
Salt-tolerance assays of *GsWRKY23*-transgenic *Arabidopsis* at the seed germination and seedling stages: (**A**) Pictorial representation of seed germination tests of WT, *atwrky23,* WT-*GsWRKY23*, and *atwrky23*-*GsWRKY23* in the presence of 0, 75, or 150 mM NaCl for 7 d. (**B**) Graphical representation of seeds’ germination rate in the presence of different NaCl concentrations. (**C**) Growth phenotype of the 3-week-old plants under 120 mM NaCl for 7 d; bar = 5 cm. (**D**) Fresh weight per plant. (**E**) Leaf chlorophyll content. (**F**) Leaf RWC. (**G**) REL value. (**H**) MDA content in the shoots. Means ± SD are shown (*n* = 3, except for fresh weight per plant, where *n* = 6, for seed germination, *n* = 32); different letters indicate groups with statistically significant differences using Duncan’s test (*p* ≤ 0.05).

**Figure 8 plants-12-03030-f008:**
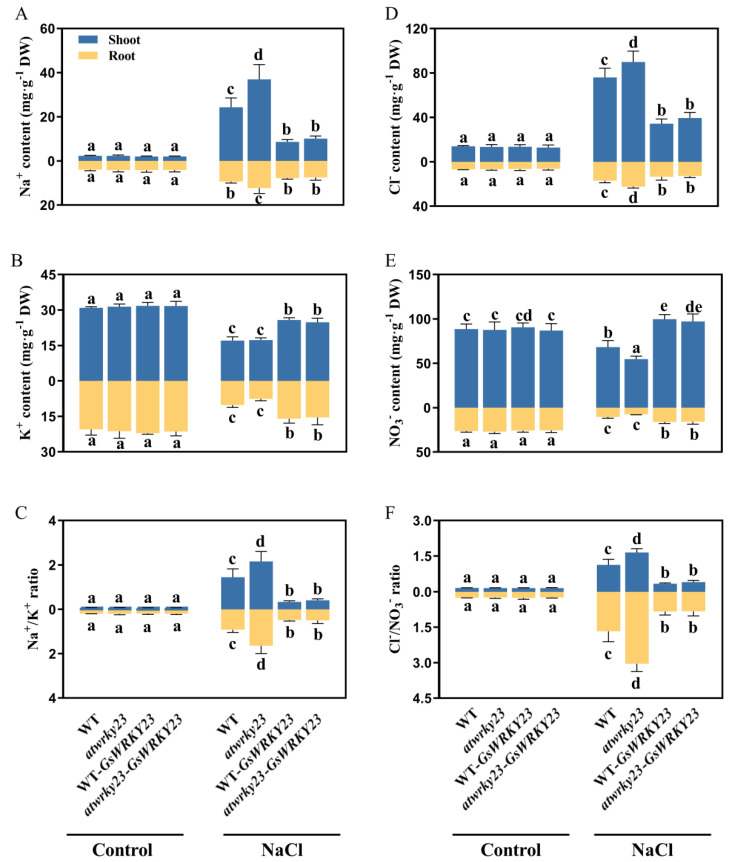
Assays of ion contents in the shoots and roots of *GsWRKY23*-transgenic *Arabidopsis* seedlings under 120 mM NaCl solutions for 7 d: (**A**) Na^+^ content, (**B**) K^+^ content, and (**C**) Na^+^/K^+^ ratio. (**D**) Cl^−^ content, (**E**) NO_3_^−^ content, and (**F**) Cl^−^/NO_3_^−^ ratio. Means ± SD are shown (*n* = 3); different letters indicate groups with statistically significant differences using Duncan’s test (*p* ≤ 0.05).

**Table 1 plants-12-03030-t001:** Differentially expressed transcription factor *WRKY* gene family enriched downstream of the MAPK4 pathway in the roots of *G. soja* BB52 seedlings under control and NaCl treatments.

Gene ID in the Transcriptome Data	Gene ID in the *G. soja* Database	Gene Names	Log_2_(Na-BB52/C-BB52)
CL12261.Contig3_All	Glysoja.06G016288	*GsWRKY6*	1.423857
CL12385.Contig2_All	Glysoja.01G000427	*GsWRKY72*	1.704426
CL13161.Contig4_All	Glysoja.05G011123	*GsWRKY72*	1.102701
CL2521.Contig3_All	Glysoja.15G040587	*GsWRKY6*	2.294813
CL4388.Contig3_All	Glysoja.03G007935	*GsWRKY42*	1.850349
CL4388.Contig6_All	Glysoja.19G052277	*GsWRKY42*	2.485088
CL4388.Contig9_All	Glysoja.03G007935	*GsWRKY42*	2.497004
CL8303.Contig1_All	Glysoja.08G022304	*GsWRKY6*	1.212138
CL8605.Contig6_All	Glysoja.18G047646	*GsWRKY33*	1.460307
Unigene11523_All	Glysoja.02G002639	*GsWRKY23*	3.917525
Unigene21237_All	Glysoja.08G020557	*GsWRKY13*	1.053174

## Data Availability

All data are available in the Appendix A or upon request.

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
