# Peer review of "Functional Characterisation of the Transcription Factor *GsWRKY23* Gene from *Glycine soja* in Overexpressed Soybean Composite Plants and *Arabidopsis* under Salt Stress"

_plants, 2023, doi:10.3390/plants12173030_

Round 1
Reviewer 1 Report (Previous Reviewer 1)
The following corrections need to be addressed before acceptance of the manuscript for publication:
(1) Present the name of the genotype/ accession used in tobacco (Nicotiana benthamiana) at Line No. 395
(2) At 4.3, there is no mention of the number of replicates used in the study. The authors need to present the detailed information in Materials and Methods
(3) At 4.8, there is no mention of the number of replicates used in the study. The authors need to present the detailed information in Materials and Methods
(4) Present the discussion with suitable sub-headings for easy understanding to the readers
Author Response
The following corrections need to be addressed before acceptance of the manuscript for publication:
- Present the name of the genotype/ accession used in tobacco (Nicotiana benthamiana) at Line No. 395
Response: Thanks for your kind suggestion, it was added and shown at Line 397.
- At 4.3, there is no mention of the number of replicates used in the study. The authors need to present the detailed information in Materials and Methods
Response: Thanks, it was revised as “Three independent biological replicates and three independent technical replicates were included for each sample.”
- At 4.8, there is no mention of the number of replicates used in the study. The authors need to present the detailed information in Materials and Methods
Response: Thanks, it was revised as “Three independent biological replicates and three independent technical replicates were included for each sample.”
(4) Present the discussion with suitable sub-headings for easy understanding to the readers
Response: Thanks, two sub-headings (3.1 ~ and 3.2 ~) was added in the revised manuscript.
Reviewer 2 Report (Previous Reviewer 3)
The authors have responded to all my concerns and improved the manuscript.
N/A
Author Response
The authors have responded to all my concerns and improved the manuscript.
(x) Minor editing of English language required
Response: Thanks. In the revised manuscript, we have made another carefull check on English language as possible as we can.
This manuscript is a resubmission of an earlier submission. The following is a list of the peer review reports and author responses from that submission.
Round 1
Reviewer 1 Report
Title: Functional Characterization of the Transcription Factor GsWRKY23 Gene from Glycine soja in Overexpressed Soybean Composite Plants and Arabidopsis under Salt Stress
Comments
1. The mean values are represented in the form of deviation from standard deviation. However, the data was recorded from samples, not the individual values of population. Here, standard error should be represented, instead of standard deviation. SE is the SD of means. The values should be indicated with respect to Sem.
2. Kindly provide the names of the species for the genes identified in homology studies of GsWRKY23.
3. On what basis the threshold similarity identified to be included as homologous. And what is the significance of this similarity study if 55.7% sequence similarity is also considered.
Reviewer 2 Report
This descriptive manuscript is the study of the a member of WRKY family in Glycine Soya and its role in salt stress response.
I have to say that I found it quite hard to read the manuscript as it requires an extensive editing of the English. Some of the experiments are OK but the way how this gene was identified and how it was linked to salt stress is highly debatable.
In more details:
Figure 1 describes that the gene was identified for its expression to Calcium signal related to plant pathogen interaction. How then this was associated to salt stress is not clear at all.
Figure 2: You need to show another gene that is not affected by the treatment as control. Similarly, you need to show a time course for the GUS lines, especially if the expression of WRK23 is not so high at 24h.
Figure 3A: I don't really understand the purpose of the yeast experiment. Rather than showing the ability of the WRKY to bind DNA sequences in general, I would rather check its ability to bind promoters of salt induced genes, checking for conserved regions in their promoters and especially check its transcriptional activity under salt.
Figure 3B: the figure legend is wrong, you cannot see eGFP in the bright field. What is the marker that you are using? It is not indicated in the figure description. Related to that the statement in line 171 is not correct. You cannot conclude that one protein works as TF just by a nuclear localisation.
Figure 4: you need a time course, especially if the gene is active in the first six hours during the treatment.
Language requires extensive editing. Many sentences are not correct (e.g. line 14, line 182).
Reviewer 3 Report
In this manusript, Sun et al., performed functional characterization of a salt-inducible transcription factor, GsWRKY23, in regulating salt tolerance by constructing soybean hairy-root composite plants and transgenic Arabidopsis. GsWRKY23 was verified as a nucleus-localized transcription factor. From the analysis of ion contents and expression of genes encoding ion transporters and channels, GsWRKY23 was proposed to enhance salt tolerance through ion homeostasis. However, there are several concerns regarding the experimental design to be addressed before publication.
1. n=3 is a small number when comparing the physiology of soybean composite plants with control. Soybean hairy-root composite plants usually have high individual variations in hairy root development. More individuals are needed to obtain reliable results.
2. Would the overexpression of GsWRKY23 also enhance the expression of SOS1 and HKT1 in Arabidopsis?
3. Is the homolog of GsWRKY23 also induced by salt in cultivated soybean? The amino acid sequence seems to be highly similar to GmWRKY3. GsWRKY23/GmWRKY3 may perform similar function in salt tolerance which worths discussion.
4. In Fig S2, why is the expression of GsWRKY23 not induced by salt stress in EV?
5. Please check the labels in Figure 3B.